# Unprecedented Neoverrucosane and Cyathane Diterpenoids with Anti-Neuroinflammatory Activity from Cultures of the Culinary-Medicinal Mushroom *Hericium erinaceus*

**DOI:** 10.3390/molecules28176380

**Published:** 2023-08-31

**Authors:** Jing Wei, Jia-yao Li, Xi-long Feng, Yilin Zhang, Xuansheng Hu, Heping Hui, Xiaodong Xue, Jianzhao Qi

**Affiliations:** 1College of Biology Pharmacy & Food Engineering, Shangluo University, Shangluo 726000, China; 2Qinba Mountains of Bio-Resource Collaborative Innovation Center of Southern Shaanxi Province, Hanzhong 723000, China; 3Shaanxi Key Laboratory of Natural Products & Chemical Biology, College of Chemistry & Pharmacy, Northwest A&F University, 3 Taicheng Road, Yangling 712100, China

**Keywords:** 13-*epi*-neoverrucosane, xylosylated cyathane diterpenoids, anti-neuroinflammatory, neurodegenerative diseases

## Abstract

The culinary medicinal mushroom *Hericium erinaceus* holds significant global esteem and has garnered heightened interest within increasingly ageing societies due to its pronounced neuroprotective and anti-neuroinflammatory properties. Within this study, two novel diterpenes, 16-carboxy-13-*epi*-neoverrucosane (**1**) and Erinacine L (**2**); three known xylosyl cyathane diterpenoids, Erinacine A (**3**), Erinacine C (**4**), and Erinacine F (**5**); and four lanostane-type triterpenoids, and three cyclic dipeptides (**10**–**12**), in addition to orcinol (**13**), were isolated from the rice-based cultivation medium of *H. erinaceus*. Their structures were determined by NMR, HR-ESI-MS, ECD, and calculated NMR. Compound **1** marks a pioneering discovery as the first verrucosane diterpene originating from basidiomycetes, amplifying the scope of fungal natural product chemistry, and the intricate stereochemistry of Compound **5** has been comprehensively assessed for the first time. Compounds **2**–**5** not only showed encouraging neurotrophic activity in rat adrenal pheochromocytoma PC-12 cells, but also significantly inhibited lipopolysaccharide (LPS)-induced nitric oxide (NO) production in BV2 microglia cell cultures with IC_50_ values as low as 5.82 ± 0.18 μM. To elucidate the mechanistic underpinnings of these bioactivities, molecular docking simulation was used to analyze and support the interaction of **1** and **2** with inducible NO synthase (iNOS), respectively. In particular, compound **2**, a cyathane-xyloside containing an unconventional hemiacetal moiety, is a compelling candidate for the prevention of neurodegenerative diseases. In summation, this investigation contributes substantively to the panorama of fungal diterpene structural diversity, concurrently furnishing additional empirical substantiation for the role of cyathane diterpenes in the amelioration of neurodegenerative afflictions.

## 1. Introduction

Mushrooms, predominantly the fruiting bodies of basidiomycetes, are an important source of natural products [1]. Mushrooms produce structurally diverse bioactive compounds that play essential roles in natural medicines, complementary and alternative medicines, and nutraceutical functional foods [1,2,3]. Additionally, many mushrooms have nutritional value, with *H. erinaceus* being a prime example. *H. erinaceus* is one of the most common edible and medicinal mushrooms, known as “Houtou” in China, “Yamabushitake” in Japan, and Lion’s Mane in the West. It has a long history of use in traditional Chinese medicine and as a delicacy due to its proven health benefits.

In the current society of increasing aging, neurodegenerative diseases such as Alzheimer’s disease (AD) have imposed heavy societal and familial burdens. Most AD drugs only alleviate symptoms but do not halt disease progression, so developing safe and effective AD treatments remains an urgent need. Studies show *H. erinaceus*-derived compounds, such as erinacines [4,5,6], promote nerve growth factor (NGF) synthesis, have anti-neuroinflammatory and neuroprotective activities, providing a basis for its effects against neurodegeneration [7,8]. Erinacines are cyathane diterpenes with a unique 5,6,7-tricyclic skeleton, named Erinacines A through Z2 [9]. Besides *Hericium* spp., cyathane diterpenes occur in *Cyathus* [10,11,12], Sarcodon [8], and other mushrooms [13,14,15,16]. Nearly 120 have been isolated from the Nidulariaceae family of basidiomycetes [10].

Verrucosane diterpenes are a class of natural compounds with fused 3,6,6,5-tetracyclic carbon skeleton; the first verrucosane diterpene was isolated from the leafy liverwort *Mylia verrucosa* Lindb in 1978 [17,18]. Verrucosane and its analogues, such as Neoverrucosane, Homoverrucosane, *Epi*-neoverrucosane, and *Epi*-homoverrucosane diterpenoids, have been consistently reported [17,19,20,21]. The structural difference between neoverrucosane and verrucosane diterpenoids lies in the fusion position of the three-membered ring with the six-membered ring [20]. Neoverrucosane analogues were first isolated with verrucosane diterpenes from *M. verrucosa* [17,18], and since then, have only been isolated from the liverwort *Plagiochila stephensoniana* [22] and the obligatory marine bacterium *Saprospira grandis* Gross (Flexibacterales) [23]. In addition, a known neoverrucosane derivative, neoverrucosan-5*β*-ol, was reported to be derived from engineered *Escherichia coli* [24]. 13-*epi*-Neoverrucosan diterpene is the epimer of neoverrucosan with the isopropyl group in *β* configuration at C-13; this compound was first discovered in *P. stephensoniana* [22]. 13-*epi*-neoverrucosane-Type diterpenoids were also identified in the liverwort *Schistochila nobilis* [19], the liverwort *Fossombronia alaskana* [25], the liverwort *Lepicolea ochroleuca* [26], the marine sponge *Axinyssa tethyoides* [21], the liverwort *P. subinflata* [27], and the marine sponge *Hamigera tarangaensis* [28]. Few bioactivity assays showed that analogues of 13-*epi*-neoverrucosane analogues still failed to inhibit the growth of HL-60 cells at concentrations up to 50 μM [28].

In the course of our ongoing research on anti-neurodegenerative chemical constituents from macrofungi, thirteen structurally diverse natural compounds were isolated and characterized in the rice medium of the culinary-medicinal mushroom *H. erinaceus*. Among them were two novel diterpenes (**1**–**2**), and eleven known secondary metabolites, including three cyathane diterpenoids (**3**–**5**), four lanostane-type triterpenoids (**6**–**9**), three cyclic dipeptides (**10**–**12**), and a phenol analogue (**13**) (Figure 1). Compound **1**, the first fungal verrucosane, and **2**, with a rare hemiacetal, were assessed for neurotrophic and anti-neuroinflammatory activities. Compounds **1**–**5** inhibited NO production in LPS-stimulated BV-2 murine microglial cells. Compound **1** enhanced nerve growth factor-mediated neurite outgrowth in rat pheochromocytoma (PC12) cells below 10 μM. Molecular docking suggested **1**–**2** suppress iNOS. The uncommon hemiacetal **2** exhibited both neuritogenic and anti-neuroinflammatory effects, holding promise for neurodegenerative diseases. This study extends the structural diversity and bioactive potential of fungal diterpenes.

## 2. Results

### 2.1. Structure Elucidation for ***1*** and ***2***

Compound **1** was isolated as a white powder. The molecular formula of **1** was determined to be C_20_H_32_O_3_, with five degrees of unsaturation, basis on HRESIMS at *m*/*z* 343.2240 [M+Na]^+^ (Appendix A). The ^1^H NMR spectrum revealed the presence of four methyls (*δ*_H_ 1.26 (3H, d, *J* = 7.0 Hz, H3-17), 1.18 (3H, s, H3-18), 0.83 (3H, s, H3-20), 0.80 (3H, s, H3-19)), one oxymethine (*δ*_H_ 4.00 (1H, dd, *J* = 10.8, 7.3 Hz, H-5)). The ^13^C NMR data revealed 20 carbon signals corresponding to four methyls, six methylenes, six methines (one oxygenated at *δ*_C_ 71.9 (C-5), four quaternary carbons, and one carbonyl at *δ*_C_ 181.2 (C-16)) (Appendix A).

The structure of **1** was determined by further analysis involving 1D NMR, HMBC and COSY spectra. The ^1^H-^1^H-COSY and HMBC spectra of **1** allowed the identification of three partial structures, CH-5/CH2-6, CH3-17/CH-15/CH-13/CH2-12/CH2-11, and CH-13/CH-14/CH-1. The planar structure for **1** was determined from the key HMBC correlations from H-5 to C-3, C-4, C-6, C-18; from H-15 to C-12, C-13, C-14, C-17, C-16; from H-13 to C-15, C-16; from H-17 to C-13, C-15, C-16; from H-18 to C-2, C-3, C-4, C-5; from H-19 to C-1, C-6, C-7, C-8; and from H-20 to C-9, C-10, C-11, C-14. The relative configuration of **1** was determined through a NOESY experiment. The key cross-peaks of H-5/18-Me, H-5/19-Me, H-14/19-Me, H-2/H-14, H-15/20-Me, and H-15/H-1 in the NOESY spectrum indicated the relative configuration (Figure 2A and Appendix A).

Compound **1** showed a similar NMR spectroscopic profile to that of 5*β*-hydroxy-*epi*-neoverrucosane [20]. The key difference between two compounds was the replacement of the methyl at C-16 (*δ*_C_ 21.3) by carbonyl (*δ*_C_ 181.2), which was implied by HMBC correlation of H-13 (*δ*_H_ 2.33) and 17-Me (*δ*_H_ 1.26) with C-16. Meanwhile, the absolute configuration of **1** was determined combining the results of NMR calculations of the chirality of C13 and C15 (Figure 2B), as well as experimental CD and calculated CD comparisons (Figure 2C). Therefore, **1** was named 16-carboxy-13-*epi*-neoverrucosane. Its IR and UV spectra were shown on Appendix A.

Compound **2** was isolated as light yellow powder with a molecular formula of C_28_H_46_O_8_ based on its HR-ESI-MS peak at *m*/*z* 533.3091 [M+Na]^+^ (*calcd* for C_28_H_46_O_8_Na, 533.3084) (Appendix A), indicating six indices of hydrogen deficiency. The UV spectrum displayed absorption maxima at 200 and 230 nm (Appendix A). Analyses of the ^1^H and ^13^C NMR spectra (Appendix A) of **2** revealed the presence of seven methyl groups, six methylene groups (one oxygenated sp3 carbons at *δ*_C_ 60.8), ten methines (one sp2 carbon at *δ*_C_ 132.0, and seven oxygenated sp3 carbons), and five quaternary carbons (three sp2 carbons at *δ*_C_ 139.4, 139.8, and 141.4 and two quaternary sp3 carbons at *δ*_C_ 43.7 and 49.7). Comprehensive analysis of the 1D and 2D NMR data (Appendix A) of **2** resulted in the whole assignment of the proton signals to their respective carbons (Table 1). 

Analysis of the ^1^H-^1^H COSY spectrum (Appendix A) of **2** revealed the presence of six spin-coupling systems (Figure 2D). In the HMBC spectrum (Appendix A), the correlations from H-17 to C-1/C-9/C-4, H-18 to C-2, H-20 to C-3, H-8 to C-6, H-11 to C-5/C-13, and H-16 to C-14 suggested the existence of a fused 5/6/7 tricarbocyclic cyathane-type scaffold in compound **2** (Figure 2D). In addition, the HMBC correlations from H-14 and C-1′, H-1′ to C-5′, H-5′ to C-3′ suggested that the erinacine-like xyloside was located at the C-14. Further analysis of the NMR data with those of reported Erinacine Z2 suggested that both compounds share the similar structure exception an acetal [-CH(OCH_3_)_2_] moiety on **2** instated of an aldehyde in Erinacine Z2 [9,29], which was confirmed by the diagnostic HMBC correlations from H-15 to C-22/C-23 (Figure 2D). Finally, the structure with relative configuration of compound **2** was determined as shown and named Erinacine L.

The NOESY cross-peaks of H-5/H-10*β*/H-11/H-17 suggested that these protons were *β*-oriented. Accordingly, the *α*-orientation of H-14 and H-16 was substantiated by the NOESY correlation of H-14/H-16/H-10*α* (Appendix A). However, the relative configurations of the remaining stereocenters could not be firmly determined from the NMR data due to the lack of enough convincing NOESY correlations. To determine absolute configurations, quantum-chemical ECD calculations at the polarizable continuum model (PCM)/B3LYP/6-311+G (d, p) level were carried out. As a result, the calculated ECD spectra for (5*R*, 6*R*, 9*R*, 11*R*, 14*S*) showed good agreement with the measured curves for **2** (Appendix A), which allowed the establishment of **2** absolute configuration as 5*R,* 6*R*, 9*R*, 11*R*, 14*S* (Figure 2E). Its IR and UV spectra were shown on Appendix A.

Eleven known compounds were identified as erinacine A (**3**) [4], erinacine C (**4**) [4] erinacine F (**5**) [5], ergosterol (**6**) [30], 5*α*,8*α*-peroxy- (22*E*,24*R*) -ergoside-6,22 -diene 3*β*-alcohol (**7**) [31], *β*-sitosterol (**8**) [32], ergosta-4,6,8(14),22-tetraen-3-one (**9**) [33], brevianamide F (**10**) [34], *cyclo*-(*trans*-4-hydroxy*-L-*prolyl*-L-*phenylalanine) (**11**) [35], *cyclo*-(*cis*-4-hydroxy-D-prolyl*-L-*leucine) (**12**) [35], and orcinol (**13**) [36] by comparing their spectroscopic data (Appendix A) with the literature. Among them, **13** was first isolated from the genus *Hericium* Mushroom. Compounds **7**–**9** are derivatives of 6, and such compounds are common in mushrooms. 

### 2.2. Structure Revision for ***5***

Compound **5** was first reported to have been isolated from the mycelium of the *H. erinaceus* in 1994 [5], and no further isolation has been reported since. In this work, **5** was identified as erinacine F by matching one-dimensional NMR data with literature report [5]. Although the planar structure of **5** was determined to be consistent with that of erinacine E [5], its absolute configuration has not been confirmed due to the absence of a two-dimensional correlation of the xylosyl group. The chirality of the C1’ of **5** was inferred to be the S configuration based on the absolute configuration of the identified xylosyl cyathane diterpenes. The remaining four C atoms of undetermined chirality are C15, C2’, C3’, and C4’, with hydroxyl modifications. Thus, there are 16 absolute configurations of **5**, i.e., I_1–I_16. Chemical shift calculations were carried out for the 16 conformations using the correlation basis set based on density functional theory (DFT) within Gaussian 16. Then the data were disposed with the help of DP4+ application (Dataset_DP4+). The results showed that the absolute configuration of 5 is 98.88% probability of I_9, i.e., four chiral carbons with the configuration 15*S*, 2’*S*, 3’*S*, 4’*R* (Figure 3). The percentage of configuration I_16, the absolute configuration of erinacine E, is only 0.004%, which proves the reliability of this result to some extent (Figure 3).

### 2.3. Neurotrophic Activity

Compounds **1**–**9** were selected for evaluation of PC-12 cell-based neurotrophic activity, and at a concentration of 5 μM, **1**–**9** all showed NGF-dependent growth promotion of PC12 cells. Among them, **2**–**5** showed significant promotion, while **1** and **6**–**9** showed weak activity. Compared with the 12.11% promotion rate of 20 ng/mL NGF, the 5 μM concentration of **2** increased the promotion rate to 24.51%, showing the strongest promotion effect (Figure 4A). To investigate the effect of different concentrations on the promotion effect of **2**, a concentration gradient experiment for **2** was performed. At a concentration of 1 μM, the promotion rate of NGF-dependent **2** was 20.73%, and when the concentration of **2** reached 9 μM, the promotion rate of **2** increased to 28.14% (Figure 4B,C). This finding suggests that the NGF-dependent promotion of **2** has a dose-dependent effect over a certain range.

### 2.4. Anti-Neuroinflammatory Activities

To further explore anti-neurodegenerative disease activity, **1**–**9** was used to test BV2 microglia cell-based anti-neuroinflammatory activity. Preliminary toxicity testing showed that at a concentration of 40 μM, **1**–**9** showed over 80% cell viability after 24 h of incubation with BV2 cells, which was superior to that (81.06%) of the positive control quercetin (Figure 5A). 

In the anti-LPS-induced inflammation assay in BV2 cells, compounds **1**–**5** inhibited LPS-induced NO production in medium with IC_50_ values ranging from 5.82 μM to 31.44 μM, while compounds **6**–**9** showed extremely weak inhibitory activity with IC_50_ values above 40.00 μM. The inhibition of NO production by compounds **1**–**2** was significantly stronger than that of the positive control quercetin (16.00 μM) (Figure 5B and Appendix A). The promising inhibitory activity demonstrated by compound **2** led to the investigation of its influence on the expression of genes involved in the pathway of NO production. In contrast to the upregulation of P-NF-κB P65, iNOS, COX-2, and TLR4 stimulated by LPS, the addition of **2** significantly downregulated the expression of these genes. In addition, the addition of **2** weakly attenuated the expression of NF-κB P65. (Figure 5C). LPS was not significantly shown to induce NF-κB P65 and GAPDH, on which the addition of **2** weakly attenuated NF-κB-P65 expression, although it did not cause changes in GAPDH expression (Figure 5C).

### 2.5. Molecular Docking Simulation of iNOS Inhibition

To better understand the anti-neuroinflammatory activity of **1**–**2**, molecular docking simulations of **1**–**2** with iNOS were performed. The docking analysis showed that **1**–**2** binds to iNOS by occupying pockets through multiple interactions (Figure 6A,B). Compound **1** interacts directly with Gln263, Arg266, Glu377, Arg381, and Asp382 to bind to iNOS with a docking energy of −22 kcal/mol CDOCKER interaction energy (Figure 6A). Compound **2** interacts directly with Met120, Gln263, Glu377, Arg381, and Trp463, and binds to iNOS with a docking energy of −37 kcal/mol CDOCKER interaction energy (Figure 6B). Compound **2** has a stronger inhibitory activity as reflected by its lower docking energy value. Both putative models share residues Gln263, Glu377, and Arg381, which may be key residues in the inhibition of iNOS activity.

Furthermore, the stability of the iNOS-**2** docking complex was evaluated using Molecular dynamics (MD) simulations. The results show that the root mean square deviation (RMSD) values tend to stabilize on a time scale of 20 ns and at 300 K (Figure 6C). The root-mean-square fluctuation (RMSF) plots of the iNOS-**2** docking complex calculated in the MD simulations at 300 K show that some of the residues show less stability. The RMSF plots of the iNOS residues calculated in the MD simulation at 300 K showed a small region with small fluctuations (Figure 6D). Visualization by PyMOL reveals that the fluctuating region contains amino acids 110–118, which form a coil (Appendix A). This region is outside the region where **2** directly interacts with iNOS (Appendix A), and changes in this region do not affect the direct interaction of **2** with iNOS. These results suggest that **2** was able be to stably bind to iNOS through the above five amino acid residues and thus affect iNOS activity.

## 3. Discussion

Mushrooms have enjoyed a longstanding dual role as both sustenance and medicinal agents, with their consumption tracing back in tandem with the trajectory of human civilization. The curative attributes and therapeutic potential intrinsic to mushrooms have held paramount significance in the annals of traditional medicine, with *Hericium erinaceus* emerging as a globally acknowledged medicinal fungus. In contemporary times, the attention lavished upon H. erinaceus has been fueled by its burgeoning reputation as a prolific source of cyathane diterpenes, exhibiting singular promise in the realm of counteracting neurodegenerative ailments [7]. Although cyathane diterpene-producing mushrooms encompasses genera such as *Hericium* [7], *Cyathus* [10], *Sarcodon* [8], and various others [8], the genus *Hericium* possesses both medicinal and culinary values. Therefore, the therapeutic use of *Hericium* mushrooms for the prevention and alleviation of neurodegenerative disorders is of great significance. The bioactive compounds derived from mushrooms often show species specificity. For example, des-A-ergostane-type compounds with anti-cancer activity, blazeispirols, are unique to *Agaricus* species [37,38,39]. *Ganoderma* acids, lanosterol-type triterpene derivatives, are the most biologically active components of the *Ganoderma* species [40]. However, the realm of cyathane diterpenoids, renowned for their potent neuroprotective attributes, traverses beyond the confines of the Hericium genus [7] to encompass *Cyathus* species [10] and other mushrooms [8]. This remarkable distribution underscores the pervasive prevalence of specific mushroom compounds. In parallel, the arena of biologically active natural pigments finds expression in styrylpyrone compounds, an illustrious group discerned widely within the fungal domains of Inonotus [2,41] and Phellinus [42] nestled within the Geomycetaceae family.

As the metabolites of cyathane diterpenoid producers continue to be elucidated, the probability of discovering new xyloside-cyathane diterpenes is decreasing. The most recent report is the discovery of eight previously undescribed cyathane-xylosides from *Dentipellis fragilis* [16]. Compound **2**, as a new xyloside-cyathane diterpene, was isolated from *H. erinaceus* together with **3** and **4**, which have been isolated several times. Compound **2** contains an unusual hemiacetal group, which is rare in cyathane diterpenes, and even in terpenoids. Although cyahookerin E [43] is the only cyathane diterpene previously found to contain a hemiacetal group, **2** is, to our knowledge, the only xyloside-cyathane diterpene currently found to contain a hemiacetal group. Compound **2** with hemiacetal group has excellent neurotrophic and anti-neuroinflammatory activities, suggesting that the hemiacetal moiety may be an essential pharmacophore for the treatment of neurodegenerative diseases. Using calculational NMR, the absolute configuration of **5** has been determined for the first time. Simultaneously, the anti-neuroinflammatory activity of 5 was tested for the first time [44]. Studies targeting neurotrophic and anti-neuroinflammatory activities at the level of signaling pathways are uncommon. Sarcodonin G, a naturally occurring cyathane diterpene, has been found to exert neuroprotective effects through activation of TrkB on the Trk signaling pathway [45]. It is proposed that the neurotrophic activity of compounds **2**–**5** is mediated in a similar manner. Another interesting compound is **13**, which is closely related to lichens. Although several chlorinated and oxidized derivatives of **13** have been found in *Hericium* species [46,47,48], no **13** has been reported from *H. erinaceus*. The present work reports, for the first time, **13** isolated from *H. erinaceus*. 

Verrucosane diterpenes are a group of terpenoids with a 3,6,6,5 tetracyclic carbon skeleton, whose 6,5-biring portion is highly congruent with that of the cyathane diterpene skeleton. The 5,6,7-tricyclic cyathane diterpene skeleton was proposed to be a precursor structure for the biosynthesis of verrucosane diterpenes [49], which was partially confirmed by a study based on transition state energy calculations [50]. Compound **1** identified in this work is the first example of an isopropyl group being oxidized to form a carboxyl group in a verrucosane diterpene, and oxidation at this position in the cyathane diterpene is only seen in cyathin C5 [11]. Verrucosane diterpenes have previously been reported to occur in lower plants [19,25], marine sponges [21,28], and bacteria [51], and here, we isolate the first such compounds from a basidiomycete. Furthermore, this is the first time that both neoverrucosane (**1**) and cyathane (**2**–**5**) diterpenes, two different types of diterpenoids, have been found in the same species. Compound **1** was found to have favorable anti-neurodegenerative disease activity, which is the first evaluation of verrucosane diterpenes for neuroprotective and anti-neuroinflammatory related activities. 

## 4. Materials and Methods

### 4.1. General Experimental Procedures

NMR spectra were recorded on a Bruker Avance III 500 MHz NMR spectrometer at 400 MHz (^1^H) and 100 MHz (^13^C) using TMS as internal standard and chemical shifts were recorded in parts per million *δ* (ppm). HR-ESI-MS was performed on an AB Sciex TripleTOF^®^ 6600 LC/MS system. Circular dichroism and IR were recorded on a Chirascan™ CD spectrometer. Column chromatography was performed using silica gel (100–200 mesh and 300–400 mesh), Sephadex LH-20 (GE Healthcare, Chicago, IL, USA), and reversed phase C18 silica gel (RP-18, GE Healthcare). For thin-layer chromatography, precoated silica gel 60 F254 plates were used and spots were visualized under UV light (210, 254 and 365 nm) or by spraying with vanilin-H_2_SO_4_ 10% solution and heating for two min. Concentration was performed with a Büchi Rotavapor R-101.

### 4.2. Fungal Material

The fungus *H. erinaceus* was obtained from the China General Microbiological Culture Collection Center (CGMCC) with the accession number CGMCC 5.579. A copy (No. HE2019) was deposited at the College of Biology Pharmacy & Food Engineering, Shangluo University, Shaanxi, China. The culture medium consisted of glucose 2%, yeast extract 0.2%, peptone 0.5%, MgSO_4_ 0.05%, and KH_2_PO_4_ 0.1%. Fermentation was carried out on a shaker at 130 rpm for 30 days at room temperature.

### 4.3. Extraction, Isolation and Purification

A total of 10 kg of rice was divided into 100 500-mL shaking flasks, each containing 100 g of rice. Then, 60 mL of sterile water was added and soaked for 2 h and finally autoclaved to complete the medium preparation. Fermentation was carried out at room temperature for 30 days after the strains were added to the medium. The culture was collected and extracted with ethyl acetate (10 L × 3). The EtOAc extract was concentrated under reduced pressure to give a crude extract (102.2 g).

The crude extract was applied to a silica gel column eluted with a gradient of CHCl_3_-MeOH (100:1, 200 mL, 50:1, 200 mL, 25:1, 200 mL, 10:1, 200 mL, 5:1, 200 mL, and MeOH 100 mL) to give six fractions (Fr. F1–F6). F-2 was separated by RP-18 (MeOH-H_2_O, 10–100%) to give five fractions (F-2-1-F-2-5). F-2-2 was purified by silica gel CC (CHCl_3_-MeOH, 20:1) followed by Sephadex LH-20 (MeOH) to afford compound **12** (13.5 mg); F-2-3 was purified by silica gel CC (CHCl_3_-MeOH, 25:1) followed by Sephadex LH-20 (MeOH) to obtain four Fractions (F-2-3-1 and F-2-3-4). F-2-3-1 was purified by semipreparative HPLC (30%, MeOH-H_2_O, 2 mL/min) to give compound **10** (*t_R_* = 17 min, 14.0 mg) and F-2-3-2 was purified by semi-preparative HPLC (40%, MeOH-H_2_O, 2 mL/min) to yield compound **13** (*t_R_* = 12 min, 11.0 mg); F-2-3-3 was purified by semi-preparative HPLC (25%, MeOH-H_2_O, 2 mL/min) to give compound **11** (*t_R_* = 20 min, 14.0 mg); F-3 was separated by RP-18 (MeOH-H_2_O, 10–100%) to give five fractions (F-3-1-F-3-5). Fraction F-3-2 was subjected to silica gel CC (petroleum ether-EtOAc, 6:4) to afford compound **7** (10.0 mg); Fraction F-3-3 was subjected to Sephadex LH-20 (MeOH) and silica gel CC (petroleum ether-EtOAc, 6:4) and further purified by semi-preparative HPLC (40%, MeOH-H_2_O, 2 mL/min) to give compound **1** (*t_R_* = 11 min, 7.0 mg). Fraction F-3-4 was subjected to Sephadex LH-20 (MeOH) and silica gel CC (petroleum ether-EtOAc, 7:3) to afford compound **6** (10.5 mg), compound **8** (7.5 mg), and compound **9** (11.4 mg); F-4 was separated by RP-18 (10–100%, MeOH-H_2_O) to obtain five subfractions (F-4-1-F-4-5). F-4-4 was purified by silica gel CC (CHCl_3_-MeOH, 25:1) followed by Sephadex LH-20 (MeOH) to obtain three Fractions (F-4-4-1, and F-4-4-3). F-4-4-1 was purified by semipreparative HPLC (15%, MeOH-H_2_O, 2 mL/min) to give compound **2** (*t_R_* = 15 min, 14.0 mg) and F-4-4-2 was purified by semi-preparative HPLC (30%, MeOH-H_2_O, 2 mL/min) to yield compound **3** (*t_R_* = 19 min, 14.0 mg). Fraction F-4-4-3 was applied to silica gel CC (petroleum ether-acetone, 4:1) to afford compound **4** (8.5 mg) and compound **5** (10.4 mg).

### 4.4. Structural Identification and Quantum Chemistry Calculations

The theoretical ECD spectra of compounds **1**–**2** were calculated using Gaussian 16 software (Gaussian Inc., Wallingford, CT, USA) based on the relative configurations determined from the NOESY spectra. Conformational searches were performed using the Monte Carlo Molecular Mechanics (MCMM) method and the OPLS_2005 force field within an energy window of 21 kJ/mol. The resulting conformers were optimized at the B3LYP/6-311+G (d, p) level of theory in Gaussian 16. ECD calculations on the optimized conformers employed B3LYP/6-311+G (d, p) in Gaussian 16, incorporating methanol solvent effects with the PCM. Boltzmann averaged ECD spectra were generated in Microsoft Excel. 

NMR chemical shift calculations were performed using DFT in Gaussian 16. Preliminary conformational searches were conducted in Spartan 14, with conformers viewed in GaussView 9.0 to prepare Gaussian input files. Geometries were optimized at the B3LYP/6-311+G (d, p) level, then stable conformations from B3LYP/6-31G (d, p) underwent magnetic shielding constant calculations at B3LYP/6-311++G (2d, p) level by solvent of methanol using PCM. The results of the previous NMR calculations were imported into Multiwfn 3.7 [52] to obtain the shielding values in different conformations. The shielding values are weighted and averaged by calculating the Boltzmann distribution from the energies obtained by conformational optimization and finally converted to chemical shifts by scaling. DP4+ [53] probability analysis compares experimental and calculated shifts for stereochemical assignment.

### 4.5. Spectroscopic Data

#### 4.5.1. Novel Compounds

16-carboxy-13-*epi*-neoverrucosane (**1**): White powder, UV(MeOH)_λmax_ (logε): 200 (3.01) nm; IR(MeOH)_υmax_: 3420, 2924, 1700, 1530, 1453, 1384, 1316 cm^−1^; ^1^H and ^13^C NMR data, Table 1; HRESI-MS *m*/*z*: 343.2240 [M+Na]^+^ (*calcd* for C_20_H_32_O_3_Na, 343.2249 (Appendix A).

Erinacine L (**2**): White powder, UV(MeOH) _λmax_ (logε): 230 (2.90) nm; IR(MeOH)_υmax_: 3364, 2928, 2888, 1689, 1455, 1379, 1241 1194 cm^−1^; ^1^H and ^13^C NMR da-ta, Table 1; HRESI-MS *m*/*z*: 533.3099 [M+Na]^+^ (*calcd* for C_28_H_46_O_8_Na, 533.3090 (Appendix A).

#### 4.5.2. Known Compounds

Erinacine A (**3**): White powder, C_25_H_36_O_6_, HRESI-MS *m*/*z*: 433.2502 [M+H]^+^. ^1^H-NMR(500 MHz, CDCl_3_) *δ*: 9.39 (1H, s, H-15), 6.79 (1H, d, *J* = 8.1 Hz, H-11), 5.89 (1H, d, *J* = 8.1 Hz, H-10), 4.68 (1H, d, *J* = 3.1 Hz, H-1′), 3.85 (1H, d, *J* = 12.4 Hz, H-5′a), 3.67 (1H, m, H-14), 3.64 (1H, m, H-4′), 3.54 (1H, m, H-3′), 3.42 (1H, m, H-2′), 3.39 (1H, m, H-13a), 3.28 (1H, dd, *J* = 18.3, 5.9 Hz, H-5′b), 2.81 (1H, m, H-18), 2.55 (1H, m, H-13b), 2.40 (1H, m, H-7a), 2.38 (2H, m, H-2), 1.75 (1H, m, H-1a), 1.65 (2H, m, H-8), 1.62 (1H, m, H-1b), 1.32 (1H, m, H-7b), 1.03 (3H, d, *J* = 6.7 Hz, H-19), 0.98 (6H, s, H-16 or H-17), 0.96 (3H, d, *J* = 6.7 Hz, H-20). ^13^C-NMR(125 MHz, CDCl_3_) *δ*: 194.5 (C-1), 154.3 (C-5), 146.5 (C-3), 145.5 (C-11), 141.5 (C-4), 138.5 (C-12), 119.9 (C-10), 104.4 (C-1′), 84.2 (C-14), 70.9 (C-3′), 70.2 (C-2′), 69.5 (C-4′), 62.4 (C-5′), 49.2 (C-9), 48.2 (C-6), 38.3 (C-1), 36.5 (C-8), 33.8 (C-7), 29.1 (C-2), 27.9 (C-13), 27.1 (C-18), 26.4 (C-16), 24.0 (C-17), 21.6 (C-19, 20), (Appendix A). It was identified by comparing its NMR data with those of Erinacine A [4].

Erinacine C (**4**): White powder, C_25_H_38_O_6_, HRESI-MS *m*/*z*: 457.2506 [M+Na]^+^. ^1^H-NMR (500 MHz, CDCl_3_) *δ*: 6.00 (1H, m, H-11), 4.80 (1H, d, *J* = 8.9 Hz, H-13), 4.56 (1H, d, *J* = 8.5 Hz, H-1′), 4.27 (1H, d, *J* = 11.9 Hz, H-15a), 4.02 (1H, m, H-15b), 3.99 (1H, m, H-5′), 3.88 (1H, m, H-14), 3.62 (1H, m, H-4′), 3.53 (1H, m, H-3′), 3.35 (1H, m, H-2′), 3.30 (1H, m, H-5′), 2.73 (1H, m, H-18), 2.46 (1H, m, H-10a), 2.35 (1H, m, H-10b), 2.29 (1H, m, H-5), 2.26 (2H, m, H-2), 1.63 (2H, m, H-7), 1.56 (1H, m, H-1a), 1.50 (1H, m, H-1b), 1.46 (2H, m, H-8), 0.99 (3H, s, H-17), 0.97 (3H, s, H-16), 0.93 (6H, m, H-19 and H-20). ^13^C-NMR(125 MHz, CDCl_3_) *δ*: 139.7 (C-3), 139.5 (C-12), 136.8 (C-4), 135.5 (C-11), 98.7 (C-1′), 79.7 (C-14), 74.5 (C-3′), 73.2 (C-13), 71.1 (C-4′), 69.9 (C-2′), 66.9 (C-5′), 65.9 (C-15), 49.4 (C-9), 42.6 (C-5), 41.5 (C-6), 38.1 (C-1), 36.5 (C-8), 28.4 (C-10), 28.3 (C-2), 27.8 (C-7), 27.0 (C-18), 24.6 (C-17), 21.9 (C-20), 21.4 (C-19), 16.6 (C-16), (Appendix A). It was identified by comparing its NMR data with those of Erinacine C [4].

Erinacine F (**5**). White powder, C_25_H_36_O_6_, HRESI-MS *m*/*z*: 433.2597 [M+H]^+^. ^1^H-NMR(500 MHz, CD_3_OD) *δ*: 5.51 (1H, m, H-11), 5.05 (1H, m, H-1′), 4.73 (1H, m, H-15), 4.26 (1H, m, H-14), 3.77 (1H, m, H-3′), 3.70 (2H, m, H-5′), 3.15 (1H, d, *J* = 10.4 Hz, H-13), 2.90 (1H, d, m, H-18), 2.70 (1H, m, H-5), 2.57 (1H, m, H-10b), 2.41 (1H, m, H-10a), 2.31 (2H, m, H-2), 1.80 (1H, m, H-7b), 1.62 (2H, m, H-1), 1.58 (1H, m, H-8b), 1.51 (1H, m, H-8a), 1.41 (1H, m, H-7a), 1.09 (3H, s, H-17), 1.01 (3H, d, *J* = 6.6 Hz, H-19 or 20), 0.99 (3H, d, *J* = 6.6 Hz, H-19 or 20), 0.98 (3H, s, H-16). ^13^C-NMR(125 MHz, CD_3_OD) *δ*: 140.2 (C-12), 140.1 (C-3), 139.3 (C-4), 121.8 (C-11), 108.4 (C-1′), 91.6 (C-14), 85.9 (C-15), 82.8 (C-2′), 79.3 (C-4′), 71.0 (C-3′), 64.3 (C-5′), 53.2 (C-13), 50.9 (C-9), 44.8 (C-5), 41.9 (C-6), 39.2 (C-1), 37.3 (C-8), 30.5 (C-10), 29.0 (C-2), 28.4 (C-7), 28.3 (C-18), 25.1 (C-17), 22.2 (C-19), 21.8 (C-20), 17.7 (C-16), (Appendix A). It was identified by comparing its NMR data with those of Erinacine F [5].

Ergosterol (**6**): White powder, C_28_H_44_O, ESI-MS *m*/*z*: 397.04 [M+H]^+^. ^1^H-NMR (500 MHz, CDCl_3_) *δ*: 5.56 (1H, m, H-6), 5.39 (1H, m, H-7), 5.22 (1H, m, H-23), 5.18 (1H, m, H-22), 3.63 (1H, m, H-3), 1.03 (3H, d, *J* = 6.6 Hz, H-21), 0.94 (3H, s, H-19), 0.91 (3H, d, *J* = 6.8 Hz, H-28), 0.84 (6H, d, *J* = 6.6 Hz, H-27 or H-26), 0.63 (3H, s, H-18). ^13^C-NMR(125 MHz, CDCl3) *δ*: 141.4 (C-5), 139.9 (C-8), 135.7 (C-22), 132.1 (C-23), 119.7 (C-6), 116.4 (C-7), 70.6 (C-3), 55.9 (C-17), 54.7 (C-14), 46.4 (C-9), 43.0 (C-24, C-13), 40.9 (C-4), 40.5 (C-20), 39.2 (C-16), 38.5 (C-1), 37.1 (C-10), 33.2 (C-25), 32.1 (C-2), 28.4 (C-12), 23.1 (C-27), 21.3 (C-11), 21.2 (C-15), 20.0 (C-26), 19.8 (C-21), 17.7 (C-28), 16.4 (C-19), 12.2 (C-18), (Appendix A). It was identified by comparing its NMR data with those of Ergosterol [30].

5*α*,8*α*-peroxy-(22*E*,24*R*)-ergoside-6,22-diene 3*β*-alcohol (**7**). White powder, C28H44O3, ESI-MS *m*/*z*: 429.03 [M+H]^+^. ^1^H-NMR (500 MHz, CD3OD) *δ*: 6.55 (1H, d, *J* = 8.5 Hz, H-6), 6.27 (1H, d, *J* = 8.5 Hz, H-7), 5.23 (1H, m, H-23), 5.21 (1H, m, H-22), 3.78 (1H, m, H-3), 1.03 (3H, d, *J* = 6.6 Hz, H-21), 0.95 (3H, d, *J* = 6.8 Hz, H-28), 0.91 (3H, s, H-19), 0.87 (9H, m, H-18, 26, 27). ^13^C-NMR(125 MHz, CDCl3) *δ*: 136.8 (C-6, 22), 133.4 (C-23), 131.7 (C-7), 83.4 (C-8), 80.7 (C-5), 66.9 (C-3), 57.5 (C-17), 53.1 (C-14), 52.7 (C-9), 45.7 (C-13), 44.3 (C-24), 41.1 (C-20), 40.6 (C-12), 38.1 (C-10), 37.7 (C-4), 35.9 (C-1), 34.3 (C-25), 30.8 (C-2), 29.7 (C-16), 24.4 (C-11), 21.5 (C-15), 21.3 (C-21), 20.4 (C-26), 20.0 (C-27), 18.6 (C-19), 18.1 (C-28), 13.2 (C-18), (Appendix A). It was identified by comparing its NMR data with those of 5*α*,8*α*-peroxy-(22*E*,24*R*)-ergoside-6,22-diene 3*β*-alcohol [31].

*β*-sitosterol (**8**): White powder, C_29_H_50_O, ESI-MS *m*/*z*: 414.99 [M+H]^+^. ^1^H-NMR (500 MHz, CDCl_3_) *δ*: 5.34 (1H, m, H-6), 3.51 (1H, m, H-3), 1.00 (3H, s, H-19), 0.92 (3H, d, *J* = 6.4 Hz, H-21), 0.86 (3H, m, H-29), 0.84 (6H, d, *J* = 7.3 Hz, H-27 or H-26), 0.67 (3H, s, H-18). ^13^C-NMR(125 MHz, CDCl_3_) *δ*: 140.9 (C-5), 121.8 (C-6), 71.9 (C-3), 56.9 (C-14), 56.2 (C-17), 50.3 (C-9), 46.0 (C-24), 42.5 (C-4), 42.4 (C-13), 39.9 (C-12), 37.4 (C-1), 36.6 (C-10), 36.2 (C-20), 34.1 (C-22), 32.1 (C-8), 32.0 (C-7), 31.8 (C-2), 29.3 (C-25), 28.4 (C-16), 26.3 (C-23), 24.4 (C-15), 23.2 (C-28), 21.2 (C-11), 19.9 (C-26), 19.5 (C-27), 19.2 (C-19), 18.9 (C-21), 12.1 (C-29), 12.0 (C-11), (Appendix A). It was identified by comparing its NMR data with those of *β*-sitosterol [32].

Ergosta-4,6,8(14),22-tetraen-3-one (**9**): White powder, C_28_H_40_O, HRESI-MS *m*/*z*: 393.3150 [M+H]^+^. ^1^H-NMR (500 MHz, CDCl_3_) *δ*: 6.59 (1H, d, *J* = 9.5 Hz, H-7), 6.01 (1H, d, *J* = 9.5 Hz, H-6), 5.70 (1H, s, H-4), 5.21 (1H, m, H-22), 5.20 (1H, m, H-23), 1.04 (3H, d, *J* = 6.7 Hz, H-21), 0.97 (3H, s, H-19), 0.94 (3H, s, H-18), 0.92 (3H, d, *J* = 6.9 Hz, H-28), 0.83 (3H, d, *J* = 6.7 Hz, H-27), 0.82 (3H, d, *J* = 6.7 Hz, H-26). ^13^C-NMR(125 MHz, CDCl_3_) *δ*: 199.5 (C-3), 164.4 (C-8), 156.1 (C-14), 135.0 (C-22), 134.1 (C-7), 132.6 (C-23), 124.5 (C-5), 124.4 (C-6), 123.0 (C-4), 55.7 (C-17), 44.3 (C-9), 44.0 (C-13), 42.9 (C-24), 39.3 (C-20), 36.8 (C-10), 35.6 (C-12), 34.2 (C-1), 34.1 (C-2), 33.1 (C-25), 27.8 (C-16), 25.4 (C-15), 21.3 (C-21), 20.0 (C-26), 19.7 (C-27), 19.1 (C-18), 19.0 (C-11), 17.7 (C-28), 16.7 (C-19), (Appendix A). It was identified by comparing its NMR data with those of ergosta-4,6,8(14),22-tetraen-3-one [33].

Brevianamide F (**10**): White powder, C_16_H_17_N_3_O_2_, ESI-MS *m*/*z*: 306.44 [M+Na]^+^. ^1^H-NMR(500 MHz, CD3OD)*δ*:7.56 (1H, d, *J* = 7.9 Hz, H-5), 7.32 (1H, d, *J* = 7.9 Hz, H-8), 7.10 (1H, m, H-2), 7.07 (1H, m, H-7), 7.03 (1H, m, H-6), 4.40 (1H, t, *J* = 4.5 Hz, H-11), 3.98 (1H, ddd, *J* = 10.2, 4.6, 0.9 Hz, H-14), 3.44 (1H, m, H-17a), 3.32 (1H, m, H-10a), 3.28 (1H, m, H-10b), 3.26 (1H, m, H-17b), 1.96 (1H, m, H-19a), 1.66 (1H, m, H-18a), 1.48 (1H, m, H-18b), 0.93 (1H, m, H-19b). ^13^C-NMR (125 MHz, CD_3_OD) *δ*: 170.7(C-13), 167.4(C-16), 138.0(C-9), 128.7(C-4), 125.5(C-2), 122.5(C-7), 119.9(C-6), 119.7(C-5), 112.2(C-8) 109.4(C-3), 60.0(C-14), 57.2(C-11), 45.9(C-17), 29.2(C-10), 29.1(C-19), 22.5(C-18) (Appendix A). It was identified by comparing its NMR data with those of Brevianamide F [34].

*Cyclo*-*(trans*-4-hydroxy*-L-*prolyl*-L-*phenylalanine) (**11)**. Yellow Oil, C14H16N2O3, ESI-MS *m*/*z*: 283.38 [M+Na]^+^. ^1^H-NMR (500 MHz, CD_3_OD) *δ*: 7.28 (5H, m, H-2′, 3′, 4′, 5′, 6′), 4.49 (1H, t, *J* = 5.0 Hz, H-3), 4.37 (1H, m, H-8), 4.28 (1H, t, *J* = 4.8 Hz, H-6), 3.73(1H, dd, *J* = 13.0, 5.1 Hz, H-9a), 3.21(1H, m, H-9b), 3.18 (2H, m, H-10), 2.06(1H, dd, *J* = 13.0, 5.9 Hz, H-7a) 1.41(1H, m, H-7b). ^13^C-NMR (125 MHz, CD3OD) *δ*: 171.3(C-5), 167.0(C-2), 137.4(C-11), 130.9(C-13, 15), 129.4(C-12, 16), 128.0(C-14), 68.5(C-8), 58.3(C-6), 57.5(C-3), 55.1(C-9), 38.8(C-10), 37.9(C-7), (Appendix A). It was identified by comparing its NMR data with those of *Cyclo*-(*trans*-4-hydroxy*-L-*prolyl*-L-*phenylalanine) [35].

*Cyclo*-(*cis*-4-hydroxy-*D*-prolyl*-L-*leucine) (**12**): Yellow Solid, C_11_H_18_N_2_O_3_, ESI-MS *m*/*z*: 227.10 [M+H]^+^. ^1^H-NMR(500 MHz, CD_3_OD)*δ*: 4.54 (1H, m, H-8), 4.47 (1H, t, *J* = 4.1 Hz, H-6), 4.17 (1H, m, H-3), 3.68(1H, dd, *J* = 12.8, 4.4 Hz, H-9a), 3.45(1H, d, *J* = 12.8 Hz, H-9b), 2.30(1H, m, H-7a), 2.09(1H, m, H-7b), 1.90(2H, m, H-10), 1.53(1H, m, H-11), 0.97(3H, d, *J* = 6.2 Hz, H-12) 0.95(3H, d, *J* = 6.2 Hz, H-13). ^13^C-NMR (125 MHz, CD_3_OD) *δ*: 173.0(C-5), 169.0(C-2), 69.1(C-8), 58.7(C-6), 55.1(C-9), 54.6(C-3), 39.3(C-10), 38.1(C-7), 25.7(C-11), 23.2(C-13), 22.1(C-12), (Appendix A). It was identified by comparing its NMR data with those of *Cyclo*-(*cis*-4-hydroxy-*D*-prolyl*-L-*leucine) [35].

Orcinol (**13**): White powder, C_7_H_8_O_2_, HRESI-MS *m*/*z*: 125.0607 [M+H]^+^. ^1^H-NMR (500 MHz, CDCl_3_) *δ*: 6.11 (2H, s, H-2, 6), 6.06 (1H, s, H-4), 2.17 (3H, s, H-7). ^13^C-NMR (125 MHz, CDCl_3_) *δ*: 159.3 (C-3, 5), 141.1 (C-1), 108.5 (C-2, 6), 100.7 (C-4), 21.5 (C-7), (Appendix A). It was identified by comparing its NMR data with those of orcinol [36].

### 4.6. Biological Activity Assay

#### 4.6.1. Evaluation of Neurotrophic Activity

The neurotrophic activity of compounds **1**–**9** was investigated using PC-12 cells obtained from the China Center for Typical Culture Collection. PC-12 cells were cultured in F-12 (Ham) medium supplemented with 10% heat-inactivated horse serum (HS), 5% heat-inactivated fetal bovine serum, 100 U/mL penicillin G, 100 μg/mL streptomycin, and 2.5 g/L sodium bicarbonate. Cells were maintained at 37 °C in a humidified 5% CO_2_ atmosphere.

As previously described, PC-12 cells were analyzed morphologically and quantified for neurite outgrowth using phase contrast microscopy. Cells were seeded in 24-well plates coated with poly*-L-*lysine at a density of 2 × 10^4^ cells/mL in standard serum medium for 24 h. Before exposure to the vehicle (0.1% dimethyl sulfoxide (DMSO)) or test compounds, the F-12 medium containing low serum (1% HS and 0.5% fetal bovine serum (FBS)) was replaced. Cells were treated with 10 μM and 20 ng/mL concentrations of test compound 1. Untreated cells served as a negative control, while 20 ng/mL nerve growth factor (NGF) was used as a positive control. Each concentration was tested in triplicate wells. After 48 h of incubation, neuronal growth of the PC-12 cells was photographed using an inverted microscope with a phase contrast objective and digital camera. Five random images were acquired from each well. The percentage of cells with neurites greater than or equal to one cell body length was considered positive for neurite outgrowth. This was quantified as a percentage of the total cells counted over the five randomly selected fields of view. The experiment was repeated at least three times and the results are presented as mean ± standard deviation.

#### 4.6.2. Evaluation of Anti-Neuroinflammatory Activity

BV-2 cells were obtained from the Cell Resource Center of Peking Union Medical College (Beijing, China). The cells were cultured in DMEM supplemented with 10% heat-inactivated FBS, 100 U/mL penicillin, and 100 μg/mL streptomycin at 37 °C under 5% CO_2_. For drug treatment, cells at 70–80% confluence in DMEM with 10% FBS were treated with 40 μM of test compounds for specified times, while control cells received an equal volume of DMSO.

Cell viability was determined by the MTT assay. BV-2 cells were seeded overnight in 96-well plates at 1.5 × 10^5^ cells/well. After 24 h of treatment with various compound concentrations, 10 μL of MTT solution (5 mg/mL) was added to each well for four hours. The supernatant was discarded, 150 μL DMSO was added to dissolve the formazan crystals, and absorbance was measured at 570 nm using a Bio-Rad plate reader. Cell viability was expressed as a percentage of the viability of DMSO-treated control cells.

For NO assays, BV2 cells were seeded at 2 × 10^5^ cells/well in 96-well plates for 24 h before treatment with 3 μg/mL LPS and various compound concentrations, using DMSO as a solvent control. Quercetin was used as a positive control drug. Equal volumes of cell culture supernatants were assayed for NO production using a Griess reaction-based kit. Briefly, 50 μL supernatant was combined with 100 μL Griess reagent in a new 96-well plate, incubated at room temperature for 15 min, and the absorbance was measured at 540 nm. Sodium nitrite standards were used construct a standard curve to calculate nitrite concentrations. The relative iNOS inhibitory activity was calculated as follows: iNOS inhibitory activity (%) = [1 − (Asp − Ab)/(Ast − Ab)] × 100%, where Asp is the absorbance of the sample reaction (containing all reagents), Ab is the absorbance of the blank group (containing all reagents without test compound), and Ast is the absorbance of the standard (containing all reagents). The IC_50_ value was defined as the concentration that reduced NO generation by 50%, and calculated via the online tool (www.aatbio.com/tools/ic50-calculator/, accessed on 6 July 2023).

### 4.7. Molecular Docking and Dynamics Simulation

To elucidate potential binding modes, we performed molecular docking simulations using the crystal structure of human iNOS (PDB: 3E7G) in Discovery Studio 2017 R2. Prior to molecular docking, the receptor proteins underwent necessary pre-treatments such as energy minimization. The receptor and ligands **1**–**2** were prepared and semi-flexible docking simulations were conducted as previously described [54,55]. The docking results were analyzed in Discovery Studio, and the 3D complexes from the docking models were visualized using PyMOL v2.5 (Schrodinger Inc., New York, NY, USA). 

MD simulations were carried out with GROMACS 2023.2 (GROMACS Development Team, Groningen, The Netherlands) using the CHARMM 36 force field. The iNOS-**2** docking complex was validated to confirm all necessary atoms were present. Topology files were generated using the pdb2gmx module, and the SPC216 water model was used to solvate the proteins. The electrically neutral system had ions added based on protein charges for MD simulations. The structures were energy minimized using 50,000 steps and a maximum force of 1000.0 kJ/mol/nm. The temperature was set to 298 K (25 °C) and system temperature was stabilized by a 100 ps NVT equilibrium, followed by a 100 ps NPT equilibrium to stabilize pressure. After equilibration, 20 ns production MD was performed for analysis. Protein stability was assessed by calculating RMSD and RMSF values and visualizing the data in Microsoft Excel 2019.

## 5. Conclusions

In conclusion, two new diterpenoids, 16-carboxy-13-*epi*-neoverrucosane (**1**) and erinacine L (**2**), together with eleven known compounds (**3**–**13**), were isolated from the metabolites of rice medium of *H. erinaceus*. Compounds **1**, **2**, and **13** were discovered for the first time from the genus, and, furthermore, **1** was isolated from the fungi for the first time. Bioactivity assay revealed that compounds **2**–**5** have good neurotrophic activity in PC-12 cells and anti-neuroinflammatory activity in in BV2 cells. Among them, **2** showed the best activity (24.51% promotion rate under 5 μM and IC_50_ of 5.82 ± 0.18 μM), probably related to the hemiacetal structure it contains. Molecular docking simulations of compounds **1** and **2** with iNOS, respectively, reflect three amino acid residues, Gln263, Glu377, and Arg381, as key sites that may influence iNOS activity. Further MD simulation confirmed the stability of the iNOS-**2** docking complex. These results enrich the structural diversity of diterpenoids derived from mushroom, and provide additional evidence for the ability of xylosylated cycloalkane diterpenes to prevent and treat neurodegenerative diseases.

## Figures and Tables

**Figure 1 molecules-28-06380-f001:**
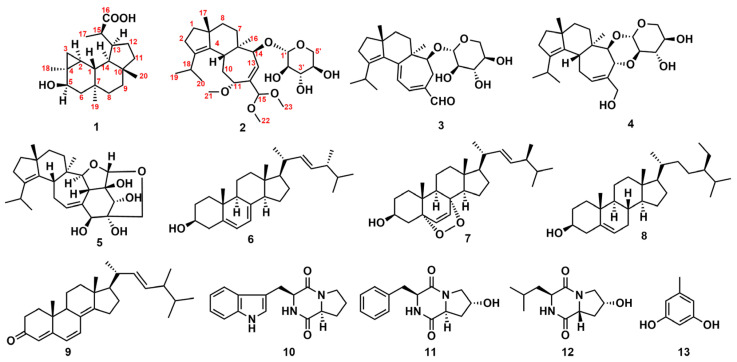
Chemical structures of compounds **1**–**13**.

**Figure 2 molecules-28-06380-f002:**
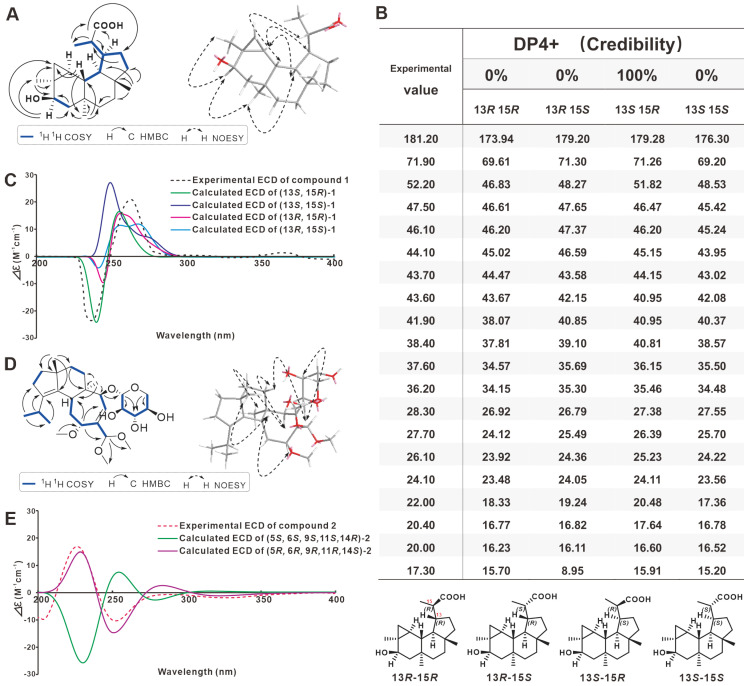
Absolute structural identification of **1** and **2**. (**A**) Key ^1^H-^1^H COSY, HMBC and NOESY correlations of **1**. (**B**,**C**) calculated and experimental NMR and ECD spectra for **1**. (**D**) Key ^1^H-^1^H COSY, HMBC and NOESY correlations of **2**. (**E**) calculated and experimental ECD spectra for **2**.

**Figure 3 molecules-28-06380-f003:**
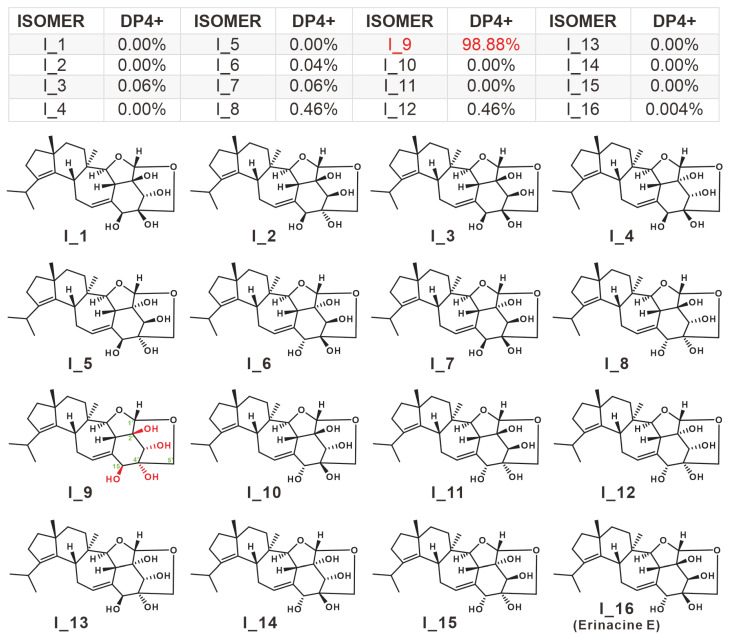
Absolute configuration identification of **5** based on calculated NMR.

**Figure 4 molecules-28-06380-f004:**
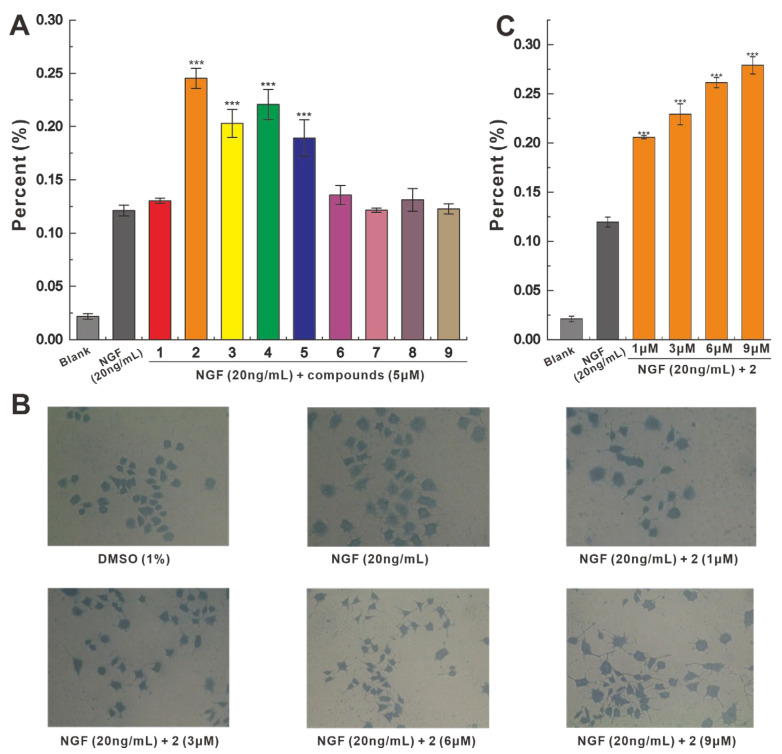
Effects of **1**–**9** on the NGF-promoted neurite outgrowth in PC-12 cells. (**A**) NGF-dependent **1**–**9** promotion rate of neurite growth in PC-12 cells. (**B**,**C**) Morphological characteristics and promotion rate of PC-12 cells treated with 2 at different concentrations. (NGF 20 ng/mL as positive control, *** *p* < 0.001).

**Figure 5 molecules-28-06380-f005:**
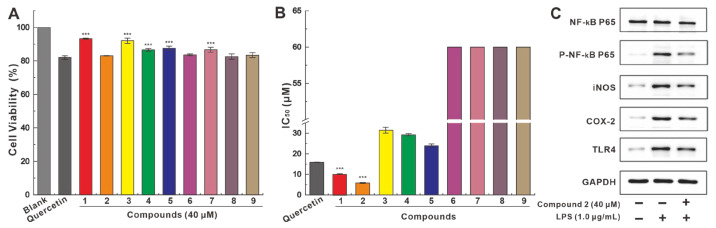
The anti-neuroinflammatory activity test of **1**–**9**. (**A**) The toxicity of **1**–**9** in BV2 microglia cells. (**B**) **1**–**9** inhibited the LPS-induced NO production in culture medium (quercetin as positive control, *** *p* < 0.001). (**C**) The effect of **2** on the expression of NF-κB P65, P-NF-κB P65, iNOS, COX-2, TLR4, and GAPDH was analyzed by western blotting.

**Figure 6 molecules-28-06380-f006:**
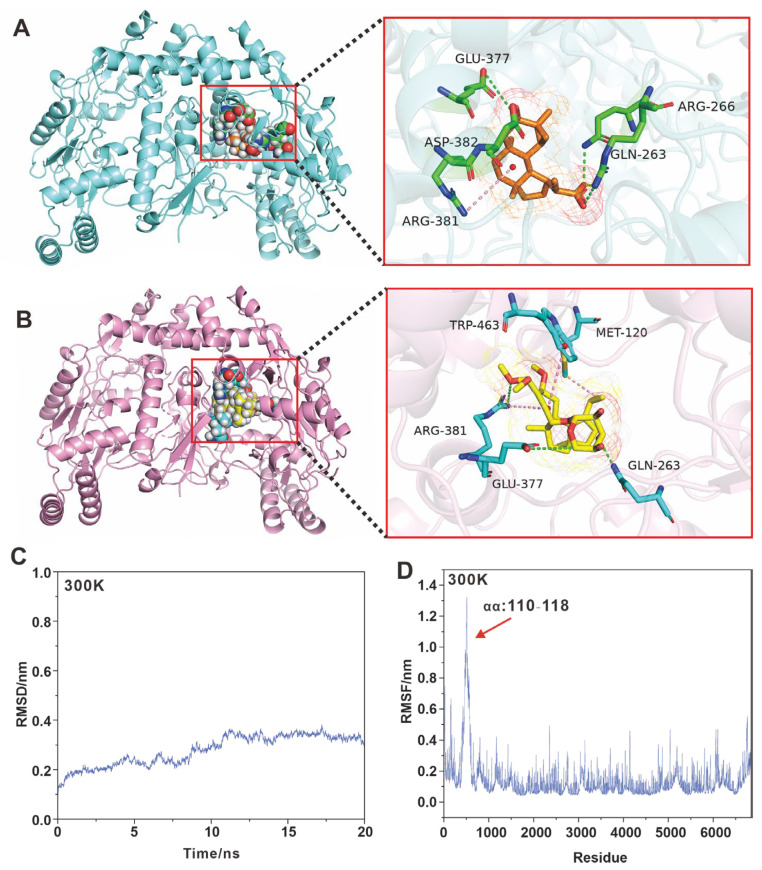
Molecular docking and molecular dynamics simulations. (**A**,**B**) Schematic representation of the molecular docking of **1** and **2** with iNOS, respectively. (**C**) MD simulates iNOS-**2** docking complex at 300 K. (**D**) RMSF plot of iNOS-**2** docking complex.

**Table 1 molecules-28-06380-t001:** The ^1^H (500 MHz) and ^13^C NMR (125 MHz) Data of **1** in CD_3_OD and **2** in CDCl_3_.

	1 (CD_3_OD)		2 (CDCl_3_)	
Position	*δ* _C_	*δ*_H_ (m, *J* in Hz)	*δ* _C_	*δ*_H_ (m, *J* in Hz)
**1**	46.1	1.29	37.1	1.52 t (7.4) 2H
**2**	28.3	0.68 m	28.6	2.30 dt (7.8,1.4) 2H
**3**	22.0	0.96 m 2H	138.6	
**4**	24.1		140.3	
**5**	71.9	4.00 dd (10.8, 7.3)	34.0	3.27 m
**6**	47.5	1.62 m H-6a 0.74 m H-6b	43.7	
**7**	38.4		34.2	1.00 td (13.4,4.2) H-7a 2.06 dt (12.9,4.8) H-7b
**8**	36.2	1.23 m H-8a 1.07 m H-8b	36.1	1.45 m 2H
**9**	37.6	1.41 m H-9a 1.32 m H-9b	49.7	
**10**	44.1		30.0	2.30 m H-10a 1.92 t (13.6) H-10b
**11**	41.9	1.50 m H-11a 1.21 m H-11b	75.1	3.88 d (5.8)
**12**	27.7	1.92 m H-12a 1.67 m H-12b	141.4	
**13**	43.6	2.35 m	132.0	6.13 d (8.0)
**14**	52.2	1.59 m	85.0	3.48 m
**15**	43.7	2.88 m	106.6	4.55 s
**16**	181.2		17.4	0.78 s
**17**	20.0	1.26 d (7.0)	24.2	1.08 s
**18**	26.1	1.18 s	27.2	2.98 p
**19**	17.3	0.80 s	21.4	0.96 d (4.6)
**20**	20.4	0.83 s	22.3	0.97 d (4.6)
**21**			57.0	3.29 s
**22**			54.7	3.36 s
**23**			53.4	3.32 s
**1′**			103.7	4.78 s
**2′**			68.5	3.80 m
**3′**			68.9	3.82 m
**4′**			70.0	3.75 m
**5′**			60.8	4.26 d (13.2) 3.47 m

## Data Availability

All data generated or analyzed during this study are included in this published article and its additional files.

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
