# Peer review of "Unprecedented Neoverrucosane and Cyathane Diterpenoids with Anti-Neuroinflammatory Activity from Cultures of the Culinary-Medicinal Mushroom Hericium erinaceus"

_molecules, 2023, doi:10.3390/molecules28176380_

Round 1
Reviewer 1 Report
The manuscript titled "Unprecedented neoverrucosane and cyathane diterpenoids with anti-neuroinflammatory activity from cultures of the culinary-medicinal mushroom Hericium erinaceus" presents a comprehensive study on the isolation, structural elucidation, and neuroprotective properties of diterpenes and other compounds from the culinary medicinal mushroom Hericium erinaceus. The authors have isolated and characterized two new diterpenes, 16-carboxyl-13-epi-neoverrucosane and Erinacine L, along with a few other compounds. The investigation into the neurotrophic and anti-neuroinflammatory activities of these compounds using cellular assays, molecular docking, and molecular dynamics simulations adds valuable insights to the potential neuroprotective properties of these compounds. Overall, the integrated approach of the work presented is interesting and proves the author's hypothesis. Besides, I would recommend some changes in the manuscript with the following comments.
1. Abstract: Can be concise and trim-down; repetitive meanings should be avoided.
2. Introduction: Some of the English terminology used is odd which needs to be updated during the revision.
3. Mention if the authors performed energy minimization of the iNOS protein structure before docking.
4. I also recommend author to extend their simulations to at least 100 ns.
5. Authors may calculate some more parameters like radius of gyration, sasa, and time evolution of hydrogen bonds from the simulated trajectory
6. The authors might consider expanding the discussion on the specific mechanisms underlying the observed neurotrophic and anti-neuroinflammatory effects. Providing more context and references to previous studies could help readers better understand the significance of the current findings.
7. To contextualize the novelty of the compounds and their activities, it could be useful to compare the newly isolated compounds with existing literature on similar compounds from other sources, and to discuss their potential advantages or differences.
8. Elaborating on the potential clinical applications of the compounds and discussing any existing therapeutic strategies for neurodegenerative diseases would add depth to the implications section.
9. What is the significance of RMSF analysis? Discuss in light of protein-ligand binding.
10. I can find some typos and grammatical mistakes throughout the manuscript which need to be corrected.
11. I also recommend the authors double-check the manuscript for abbreviations used.
I can find some typos and grammatical mistakes throughout the manuscript which need to be corrected.
Author Response
The manuscript titled "Unprecedented neoverrucosane and cyathane diterpenoids with anti-neuroinflammatory activity from cultures of the culinary-medicinal mushroom Hericium erinaceus" presents a comprehensive study on the isolation, structural elucidation, and neuroprotective properties of diterpenes and other compounds from the culinary medicinal mushroom Hericium erinaceus. The authors have isolated and characterized two new diterpenes, 16-carboxyl-13-epi-neoverrucosane and Erinacine L, along with a few other compounds. The investigation into the neurotrophic and anti-neuroinflammatory activities of these compounds using cellular assays, molecular docking, and molecular dynamics simulations adds valuable insights to the potential neuroprotective properties of these compounds. Overall, the integrated approach of the work presented is interesting and proves the author's hypothesis. Besides, I would recommend some changes in the manuscript with the following comments.
A: Thank you for your efforts in reviewing the manuscript. Your approval of the manuscript encourages us to further improve its quality.
- Abstract: Can be concise and trim-down; repetitive meanings should be avoided.
A1:Thank you for your suggestions, and we have modified the abstract in an effort to be concise.
- Introduction: Some of the English terminology used is odd which needs to be updated during the revision.
A2: Thank you for your careful review, changes have been made.
- Mention if the authors performed energy minimization of the iNOS protein structure before docking.
A3: Energy minimization of receptor proteins is a preparation for molecular docking, and we operate as such. The relevant description has been added in section 4.7.
- I also recommend author to extend their simulations to at least 100 ns.
A4: Thank you very much for your suggestion, a longer simulation time might have given better results. As you know, the crystal structure of the iNOS receptor contains nearly 500 amino acids in the monomer, and its docking complexe with compound 2 has molecular weights of 97 kDa, respectively. MD of such a relatively large system is very hardware resource intensive. In addition, a simulation time of 20 ns was effective in most studies.
- Authors may calculate some more parameters like radius of gyration, sasa, and time evolution of hydrogen bonds from the simulated trajectory.
A5: Thank you for mentioning molecular dynamics simulations. The molecular dynamics simulation technique used in this manuscript is used to support molecular docking structures aimed at exploring possible mechanisms of inhibition of iNOS by compounds. The two parameters RMSD and RMSF used in this manuscript are sufficient for the experimental requirements and have been used in similar studies.
- The authors might consider expanding the discussion on the specific mechanisms underlying the observed neurotrophic and anti-neuroinflammatory effects. Providing more context and references to previous studies could help readers better understand the significance of the current findings.
A6: We thank you for the constructive comments that were helpful in revising the manuscript, and, in our opinion, considerably improved it. A related discussion of neurotrophic and anti-neuroinflammatory effects has been added to the first paragraph of the Discussion section.
- To contextualize the novelty of the compounds and their activities, it could be useful to compare the newly isolated compounds with existing literature on similar compounds from other sources, and to discuss their potential advantage
A7: Thank you so much for your good advices. The novelty of the compounds of interest, especially the new compounds, is clearly and extensively discussed in the second paragraph of the Discussion section. For the activity of the compounds of interest, a brief discussion of activity comparisons is provided in Results section 2.3-2.5. The discussion of activity comparisons with other relevant compounds is another interesting suggestion that we will consider including in future relevant studies.
- Elaborating on the potential clinical applications of the compounds and discussing any existing therapeutic strategies for neurodegenerative diseases would add depth to the implications section.
A8: This is a good idea, but not entirely applicable to this manuscript. We have partially adopted the suggestion, and the relevant description has been added to the Discussion section.
- What is the significance of RMSF analysis? Discuss in light of protein-ligand binding.
A9:
The RMSF reflects the degree of freedom of each atom in the complex, and in this experiment the RMSF reflects the relatively high instability of amino acids 110-118 of iNOS. amino acids 110-118 do not overlap with the binding region of iNOS with Compound 2, and therefore will not affect the stability of the docking complex of iNOS with Compound 2.
- I can find some typos and grammatical mistakes throughout the manuscript which need to be corrected.
A10: The manuscript has been re-checked by a native speaker.
- I also recommend the authors double-check the manuscript for abbreviations used.
A11: Revised and thanks.
12: I can find some typos and grammatical mistakes throughout the manuscript which need to be corrected.
A12: The manuscript has been re-checked by a native speaker.
Reviewer 2 Report
The manuscript represents interesting data on two new and several known metabolites produced by Hericium erinaceus that is considered edible in Eurasia. In this research I found just minor mistakes like: compound names are sometimes given with first uppercase letter, sometime their numbers are not in bold. Please, check through the text. In the discussion section, I would mention toxicological information on the potent anti-inflammatory compounds or its absence to make their prospective for drug development clearer; please, also discuss the yield of the compounds as their biotechnological perspective. The preparation of the rice media was described unclear (4.3, lines 311-313); please.re-wright. Also pay attention on:
p.1 line 36: check words/phrase
p.2 line 77: check words/phrase
p. 4 line 108: check words/phrase
p. 7 line 191 : concentration 40 mM?
p. 7 line 202: check words/phrase
p. 11 line 368: Erinacines?
p. 14 line 542: metabolites are isolated from the fungal culture, not medium
Author Response
The manuscript represents interesting data on two new and several known metabolites produced by Hericium erinaceus that is considered edible in Eurasia.
Q1: In this research I found just minor mistakes like: compound names are sometimes given with first uppercase letter, sometime their numbers are not in bold. Please, check through the text.
A1: we have checked the full text and revised related errors.
Q2: In the discussion section, I would mention toxicological information on the potent anti-inflammatory compounds or its absence to make their prospective for drug development clearer;
A2: Prior to the anti-neuroinflammatory activity, we tested the toxicity of compounds 1-5 on BV2 cells (see Result 2.4). The analysis of the toxicological properties you mentioned is also a topic of great interest to us and will be explored in subsequent in-depth investigations of the anti-neuroinflammatory activity of such compounds.
Q3: please, also discuss the yield of the compounds as their biotechnological perspective.
A3:This consideration is an interesting topic, but not appropriate for this manuscript. Although we used rice medium to ferment the mycelium of this strain to obtain metabolites, no biotechnology was involved.
Q4:The preparation of the rice media was described unclear (4.3, lines 311-313); please.re-wright.
A4:Details have been added and the description is now complete and clear.
Q5:Also pay attention on:
p.1 line 36: check words/phrase
p.2 line 77: check words/phrase
- 4 line 108: check words/phrase
- 7 line 191: concentration 40 mM?
- 7 line 202: check words/phrase
- 11 line 368: Erinacines?
- 14 line 542: metabolites are isolated from the fungal culture, not medium,
A5:we thank you for your meticulous examination of our work. The corrections of these details were helpful in improving the quality of the manuscript.
Reviewer 3 Report
Please see the attachment

Minor editing of English language required
Author Response
Q1. Some contradictions are in Abstract and Conclusions:
A1: Confusion over compound numbering led to this error, which has been corrected.
Q2: Line 24: something is to be changed (either number or characteristics), this neoverrucosane (1) is not a cyathane-xyloside with an unusual hemiacetal group.
A2: This was also a bug where the compounds were misnumbered and has been fixed.
Q3: Line 542: "Compounds 1, 2, 13 were first discovered from the genus". However, in the Section 2.1: "Among them (among other eleven compounds apart from 1-2 – commented by a reviewer), 10-13 were first isolated from the genus Hericium" (Line 152).
Q3: Thank you for your careful reading, the two descriptions do conflict. This was caused by an error in the description at line 152, which has been corrected. Again, many thanks.
Q4: Line 544: "compounds 1-5 have good neurotrophic activity in PC-12 cells". However, in the Section 2.3 "Neurotrophic activity", it is reported that "2-5 showed significant promotion, while 1 and 6-9 showed weak activity (Line 179)."
Q4: There was an error in the statement about activity in line 544, which has been corrected.
Q5:Conclusions, Lines 545-546 "24.51% promotion rate under 5 μM and IC50 of 5.82 μM". However, the value of IC50 equal to 5.82 μM can not be found in the rest text of the manuscript. Only in the Abstract similar quantity could be seen: "... with IC50 values as low as 6.00 μM" (Line 22). Accordingly, a procedure of IC50 calculation is absent in Materials and Methods.
A5:The IC50 for compound 2 is 5.82 μM (see Figure 5B and table S1).22 The IC50 of 6.00 μM in line 22 was incorrectly described and has been corrected. The method for determining the IC50 has been added in section 4.6.2.
Q6:he section "Discussion" in its current form should be better combined with the previous section to yield "Results and Discussion". The given manuscript organized with "Discussion" entails somewhat amount of undesirable duplications of "Introduction" and "Results", and does not report new aspects compared to these sections. Alternatively, the section "Discussion" should be revised.
A6:We appreciate your thoughtful comments and suggestion. The three sections "Introduction", "Results" and "Discussion" have been revised to minimize duplication of content. At the same time, the description of discussability in the Results section has been moved to the Discussion section.
Q7: It would be useful to provide literary data on the neurotrophic activity of diterpenes structured similarly to the compounds described in the manuscript, data obtained by other researchers on the inhibition by diterpenes of the induced NO production in cell lines, as well as on the quantum chemical simulations applied previously to similar structures with iNOS.
A7: Thank you for your meaningful suggestions. The IC50 values for compounds 1-9 against iNOS are supplemented in table S1. The key data for the quantitative calculations have been shown in Figures 3 and 4.
Q7: Other proposed corrections to the manuscript
- The abbreviation for "nerve growth factor" should be done at the first appearance in the text, since "NGF" is met frequently throughout the manuscript.
- Lines 14, 112, 364, 540: "16-carboxyl-13-epi-neoverrucosane" should be replaced by "16- carboxy-13-epi-neoverrucosane".
- Lines 48-49: it would be useful to note here that Cyathus is a genus of fungi in the Nidulariaceae family of basidiomycetes.
(4) Line 51: "from Mylia verrucosa " should be replaced by "from the leafy liverwort Mylia verrucosa Lindb.", since only fungal species were mentioned in the paragraphs above.
(5) Lines 57: "from Plagiochila stephensoniana" should be replaced by "from the liverwort Plagiochila stephensoniana"
(6) Line 58: "and Saprospira grandis" should be replaced by "and the obligatory marine bacterium Saprospira grandis Gross (Flexibacterales)".
(7) Line 62-64: "identified in Schistochila nobilis [19], Fossombronia alaskana [25], Lepicolea ochroleuca [26], Axinyssa tethyoides [21], P. subinflata [27], and Hamigera tarangaensis [28]" should be replaced by "identified in the liverwort Schistochila nobilis [19], the liverwort Fossombronia alaskana [25], the liverwort Lepicolea ochroleuca [26], the marine sponge Axinyssa tethyoides [21], the liverwort Pleurozia subinflata [27], and the marine sponge Hamigera tarangaensis [28]". In Ref. 27, the Pleurozia genus is at its first appearance in the manuscript, and could not be abbreviated as P. subinflata. It would be much better to regroup the works at Lines 62-64 into sequence referring to the liverworts followed by the sequence referring to the marine sponges, or vice versa. In this case, "liverwort" and "marine sponge" are disposed of being used repeatedly.
(8) Lines 67-80 are for Conclusions or Results, if any, and Figure 1 is for Results. However, before being moved to another part of the manuscript, several inconsistencies should be corrected (Line 68 – "fourteen compounds"; Line 74 – "Compounds 1-3 inhibited nitric oxide production"; Line 76 – "Molecular docking suggested 1-3 suppress ...."; Line 77 – "The uncommon hemiacetal 1 exhibited ...").
(9) After Line 66, the formulation of the manuscript purpose in needed before the "Results".
(10) Lines 253-259: there are text duplications (with Lines 245-247) or excessive information (Lines 255-259).
A8: We appreciate and thank you for your critical review of the details. We have made careful corrections in response to your suggestions or remarks. The refinement of these details greatly improves the rigor and accuracy of the manuscript.
Thanks again for these precious and professional comments.
Q8: Lines 22-24, 547: "Molecular docking and molecular dynamics (MD) simulations were used to analyse and support the interaction of 1 and 2 with inducible nitric oxide synthase (iNOS), respectively". In this sentence, the meaning of the word "respectively" is not clear taking into account that the molecular docking simulations were performed for both (1) and (2), however, the stability of the docking complex of iNOS by means of MD was evaluated with (2). The same is for Line 547 "Molecular docking simulations of compounds 1 and 2 with iNOS, respectively, ...".
A8: Thank you for your careful review of the details, there is indeed an error in the description here. We have made changes and the description is now more objective and scientific.